# Fast and accurate *Ab Initio* Protein structure prediction using deep learning potentials

**Robin Pearce**[1], **Yang Li**[1], **Gilbert S. Omenn**[1,2], **Yang Zhang**[1,3]*

**1** Department of Computational Medicine and Bioinformatics, University of Michigan, Ann Arbor, Michigan, United States of America, **2** Departments of Internal Medicine and Human Genetics and School of Public Health, University of Michigan, Ann Arbor, Michigan, United States of America, **3** Department of Biological Chemistry, University of Michigan, Ann Arbor, Michigan, United States of America

* zhang@zhanggroup.org

## Abstract

Despite the immense progress recently witnessed in protein structure prediction, the modeling accuracy for proteins that lack sequence and/or structure homologs remains to be improved. We developed an open-source program, DeepFold, which integrates spatial restraints predicted by multi-task deep residual neural-networks along with a knowledge-based energy function to guide its gradient-descent folding simulations. The results on large-scale benchmark tests showed that DeepFold creates full-length models with accuracy significantly beyond classical folding approaches and other leading deep learning methods. Of particular interest is the modeling performance on the most difficult targets with very few homologous sequences, where DeepFold achieved an average TM-score that was 40.3% higher than trRosetta and 44.9% higher than DMPfold. Furthermore, the folding simulations for DeepFold were 262 times faster than traditional fragment assembly simulations. These results demonstrate the power of accurately predicted deep learning potentials to improve both the accuracy and speed of *ab initio* protein structure prediction.

## Author summary

Template-free protein structure prediction remains an important unsolved problem. We proposed a new pipeline to construct full-length protein structures by coupling multiple-level deep learning potentials with fast gradient-based folding simulations. The large-scale benchmark tests demonstrated significant advantages in both accuracy and speed over other fragment-assembly and deep learning-based approaches. The results revealed that the key factor for the success of the deep learning approach is its ability to provide an abundant set of accurate spatial restraints ($\sim 93^*L$ where $L$ is the protein length), which help smooth the energy landscape and make gradient-based simulation searching a feasible optimization tool. Nevertheless, extensive folding simulations are still needed for the cases where only sparse restraints are available as provided by threading alignments and low-resolution structural biology experiments.

This is a *PLOS Computational Biology* Methods paper.

**Data Availability Statement:** All relevant data are within the manuscript and its Supporting Information files.

**Funding:** This work is supported in part by the National Institute of General Medical Sciences (GM136422, S10OD026825 to YZ), the National

Institute of Allergy and Infectious Diseases
(AI134678 to YZ), the National Science Foundation
(IIS1901191, DBI2030790, MTM2025426 to YZ),
and the National Institutes of Health
(U24CA210967, P30ES017885 to GSO). The
funders had no role in study design, data collection
and analysis, decision to publish, or preparation of
the manuscript.

**Competing interests:** The authors have declared
that no competing interests exist.

## Introduction

The goal of protein structure prediction is to determine the spatial location of every atom in a protein from its primary sequence. Depending on whether reliable structural templates are available in the PDB, protein structure prediction methods have been divided into template-based modeling (TBM) and template-free (FM) approaches, the latter of which is also called *ab initio* modeling [1]. For many years, TBM has been the most reliable method for modeling protein structures; however, its accuracy is essentially determined by the availability of close homologous templates and the quality of the query-template alignments. Conversely, *ab initio* methods are designed to use advanced energy functions and sampling techniques to improve the folding performance for proteins that lack homologous templates in the PDB. However, due to the inaccuracy in force field design and the limitations of conformational search engines, the performance of the physics-based FM methods for non-homologous targets has remained significantly worse than that of the TBM methods for targets with readily identifiable homologous templates [2, 3].

Throughout the last few years, the use of deep learning techniques to predict spatial restraints from sequence and/or multiple sequence alignments (MSAs) has dramatically improved the accuracy of *ab initio* structure prediction [4]. For example, in CASP11 and CASP12, predictors primarily used direct coupling analysis from MSAs and shallow neural networks to predict contact maps, where the prediction accuracy largely relied on the identification of abundant sequence homologs in order to accurately predict contacts based on the information from correlated mutation patterns [5]. In the CASP13 experiment, however, the top-ranked server groups, Zhang-Server and QUARK, used contact maps predicted by deep convolutional residual networks (ResNets) [6] to guide the I-TASSER [7] and QUARK [8] folding simulations, respectively, which greatly improved the contact prediction and folding accuracies for the physics- and knowledge-based modeling approaches. This was especially apparent for targets that lacked homologous templates and high-quality MSAs [5]. There are also gains that can be obtained from improving protein sequence matching itself, which yields substantially more of the structure contacts [9]. Here, a contact map is specified by a binary $L \times L$ matrix, where $L$ is the protein length and each entry indicates whether the Cβ atoms (or Cα atoms for Glycine) of two residues are <8Å apart from each other. In the most recent CASP experiment, CASP14, multiple deep learning constraints, including distance maps, which are conceptually similar to contact maps but include inter-residue distance information [10, 11], inter-residue dihedral angles [12] and hydrogen-bonding networks [13], were integrated with the folding simulations. The results demonstrated significant improvements over the contact-based structure assembly approaches, due to the introduction of more precise spatial information to guide the folding simulations [13].

Despite the improvement in modeling accuracy, the approaches built on traditional fragment/template assembly folding techniques, such as I-TASSER [7], Rosetta [14] and QUARK [8], often require lengthy simulation times, especially for longer proteins, which hinders them from large-scale modeling applications. In fact, the necessity of extensive conformational sampling required for *ab initio* modeling is due to the immense structure space and complex energy landscape associated with protein folding. Although this may still be required when integrated with sparse spatial constraints (e.g., around $n^{*}L$ restraints where $n<1$) from threading alignments and low-resolution experiments [15–17], the advanced deep learning techniques can now provide abundant ($>20^{*}L$) high-quality restraints. These abundant and accurate restraints can smooth the rough protein folding energy landscape to a large degree. In this regard, extensive folding simulations may no longer be needed, which partially explains the remarkable success enjoyed by other teams in the CASP experiments such as AlphaFold

[11] in CASP13 and trRosetta [12] in CASP14, which constructed structural models using local gradient-descent based conformational searching procedures.

Inspired by these advances, we have developed a fast open-source protein folding pipeline, DeepFold, which combines a general knowledge-based statistical force field with a deep learning-based potential produced by the new DeepPotential program to improve the speed and accuracy of *ab initio* protein structure prediction. The pipeline was carefully benchmarked on large-scale datasets and showed superiority over other leading structure prediction approaches, all with greatly reduced simulation times compared to traditional folding simulation methods. Each component of the program, including the deep learning models and L-BFGS structure optimization pipeline, is integrated into an easy-to-use, stand-alone package available at both https://zhanggroup.org/DeepFold and https://github.com/robpearc/DeepFold. Meanwhile, an online webserver for DeepFold is available at https://zhanggroup.org/DeepFold, where users can apply the method to generate structure models for their own protein sequences.

## Results and discussion

### Distance and orientation restraints have the dominant impact on global fold accuracy

As shown in Fig 1, DeepFold starts by searching the query sequence through multiple whole-genome and metagenomic databases using DeepMSA2 [18] to create an MSA. Next, the co-evolutionary coupling matrices are extracted from the resulting MSA and used as input

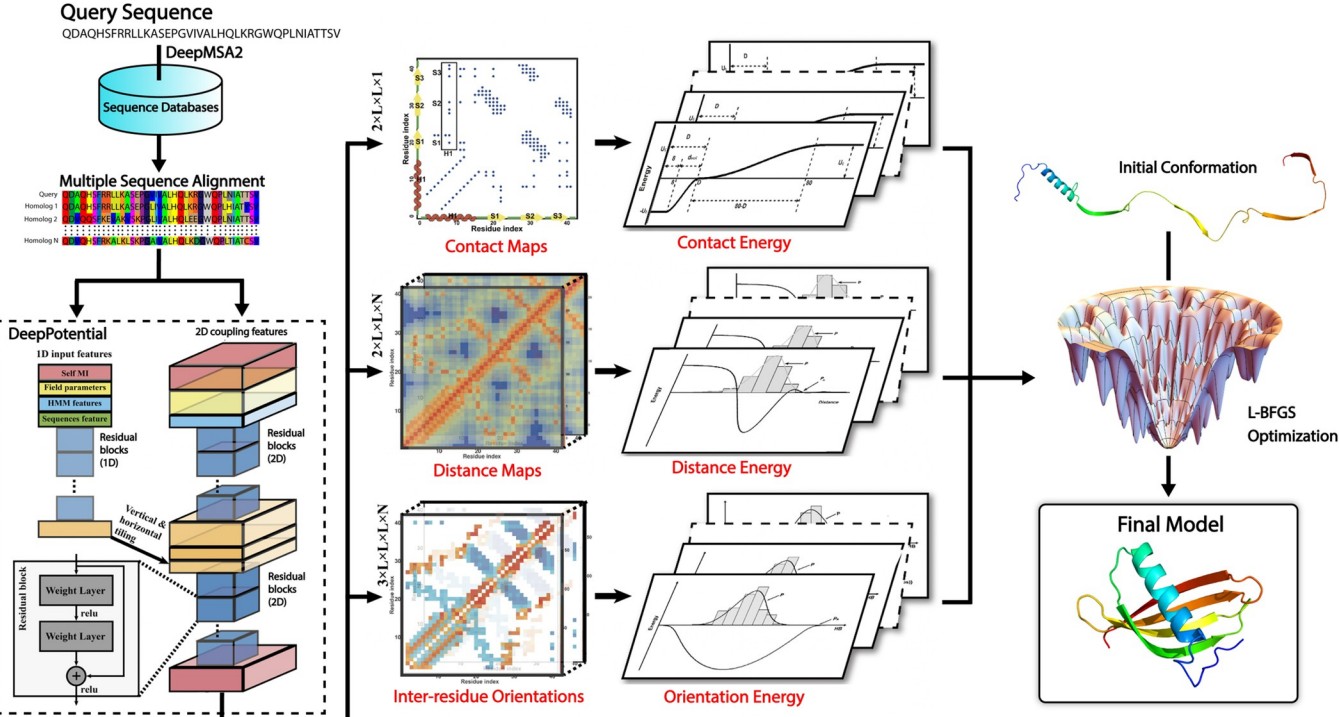

**Fig 1. Overview of the DeepFold pipeline.** Starting from a query amino acid sequence, DeepMSA2 is used to search the query against multiple whole-genome and metagenome sequence databases to create a multiple sequence alignment (MSA). The MSA is then used by DeepPotential to derive input features based on co-evolutionary analyses for the deep ResNet training. DeepPotential outputs the probability distribution of the Cβ-Cβ/Cα-Cα contact and distance maps as well as the inter-residue orientations. These restraint potentials along with the inherent statistical energy function are used to guide the L-BFGS folding simulations for final full-length structure model construction.

features by the deep ResNet architecture of DeepPotential to predict spatial restraints, including distance/contact maps and inter-residue torsion angle orientations. These restraints are then converted into a deep learning-based potential, which is used along with a general knowledge-based physical potential to guide the L-BFGS folding simulations for full-length model generation (see Methods).

To test DeepFold, we collected a set of 221 non-redundant (<30% sequence identity to each other) protein domains from the SCOPe 2.06 database and FM targets from CASP9-12. These proteins were non-homologous (with a sequence identity <30%) to the training dataset of DeepFold and were all defined as Hard threading targets by LOMETS [19] after excluding homologous templates with >30% sequence identity to the query. Here, a Hard target is a protein for which LOMETS could not identify a significant template, allowing for a systematic evaluation of the developed method on *ab initio* modeling targets. To examine the importance of the different components of the DeepFold energy function, we ran DeepFold using different combinations of spatial restraints from DeepPotential for the 221 test proteins, where the modeling results are summarized in Fig 2 and S1 Table in the Supporting Information (SI).

Overall, the baseline potential using just the general physical energy function (GE in S1 Table and Fig 2) achieved an average TM-score of only 0.184. Furthermore, when considering

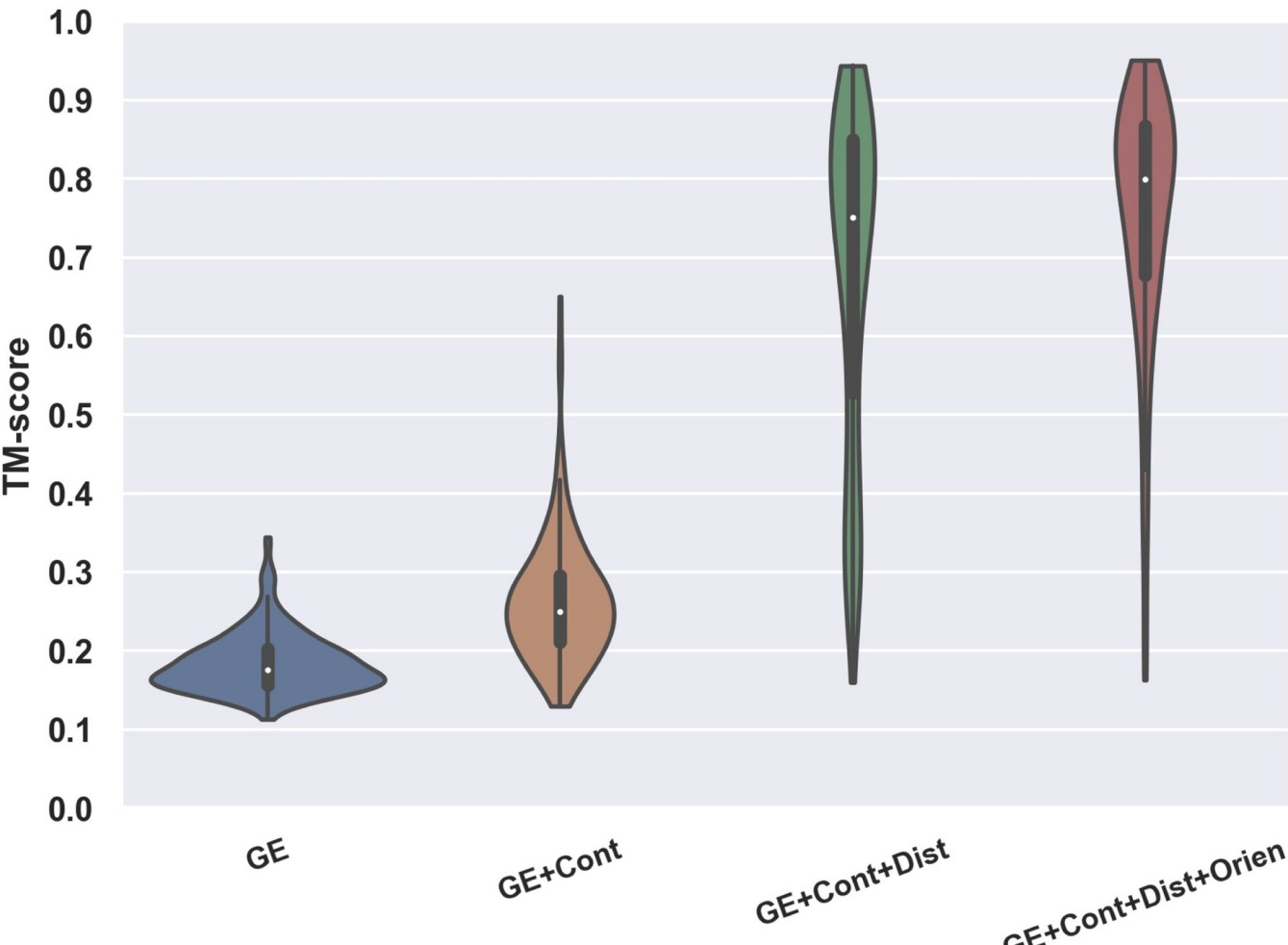

**Fig 2. Contribution of the various spatial restraints and energy terms on the DeepFold modeling accuracy, where the violin plot shows the TM-score of DeepFold using different combinations of energy terms/restraints on the 221 test proteins.**

a cutoff TM-score $\geq$0.5 to indicate a correctly folded model, which would mean the predicted model and native structure share the same global fold [8, 20], the baseline energy function was unable to correctly fold any of the test proteins (S1 Table). Given that the coupling of a similar force field with replica-exchange Monte Carlo simulations in QUARK could fold substantially more proteins with a much higher average TM-score [8], this result suggests that one major reason for the failure here is due to the frustration of the baseline energy landscape, which cannot be quickly explored by gradient-based searching methods. The further inclusion of Cα and Cβ contact restraints improved the TM-score to 0.263, where 4 of the 221 test proteins, or 1.8%, were successfully folded with TM-scores $\geq$0.5. The addition of the Cα and Cβ distance restraints dramatically improved the average TM-score on the test dataset to 0.677, representing an increase of 157.4%, where 76.0% of the test proteins were correctly folded. Lastly, the inclusion of the inter-residue orientations further improved the average TM-score to 0.751 and the percent of successfully folded proteins to 92.3%. Overall, as the level of detail in the restraints increased, the energy landscape became increasingly smooth and thus the L-BFGS folding simulations resulted in increased average TM-scores across the test proteins.

Although the addition of inter-residue distances to the energy function brought about the highest increase in accuracy, one interesting observation is the synergistic effect observed when combining different components of the restraints. For example, the addition of inter-residue orientations improved DeepFold's ability to find structures that optimally satisfied the distance restraints. As evidence of this, in S2 Table we present the mean absolute errors (MAEs) for the top $n^*L$ long-range distance restraints which were calculated between the DeepPotential predicted distance maps and the final DeepFold models with and without the use of the orientation restraints. The data in S2 Table shows that the introduction of inter-residue orientations helped to significantly decrease the MAE between the predicted distance maps and the structure models. For example, when considering the top $2^*L$ distance restraints, which were sorted by their DeepPotential distance prediction confidence scores, the MAE was 0.74 Å when DeepFold was run only using the GE and contact/distance restraints, whereas the MAE was reduced by 17.6% to 0.61 Å when the orientation restraints were added. Therefore, not only do orientations provide useful geometric information on their own, they also help further smooth the energy landscape and facilitate the L-BFGS search to identify energy basins that satisfy the ensemble of spatial restraints.

Furthermore, inter-residue orientations were particularly useful for folding β-proteins. As seen in S3 Table, the inclusion of orientations increased the average TM-score for β-proteins from 0.590 to 0.706, corresponding to a 19.7% improvement, which was significantly higher than the 10.9% improvement observed on the overall dataset (S1 Table); this makes sense intuitively given the intricate hydrogen bonding patterns present in β-proteins that would require more detailed local inter-residue dihedral angle restraint information to properly recapitulate. Fig 3A presents an illustrative example from SCOPe protein d1jqpa1, which adopts a β-barrel fold. The model built without orientations had a low TM-score of 0.313 and an RMSD of 11.43 Å, where the MAE between the top $2^*L$ DeepPotential distances and the model without orientations was 0.87 Å. In contrast, the model built using the orientation restraints had a drastically improved TM-score of 0.800 and an RMSD of 2.74 Å. Additionally, the MAE between the top $2^*L$ DeepPotential distances and the model improved to 0.61 Å. Thus, the orientation restraints provide complementary information to the distance maps and had a particularly important role for folding β-proteins.

## The general knowledge-based energy function improves local physical structure quality

The rapid improvement in the accuracy of deep learning-based restraint prediction has called into question the role of the physical energy function in the era of deep learning. Indeed, we

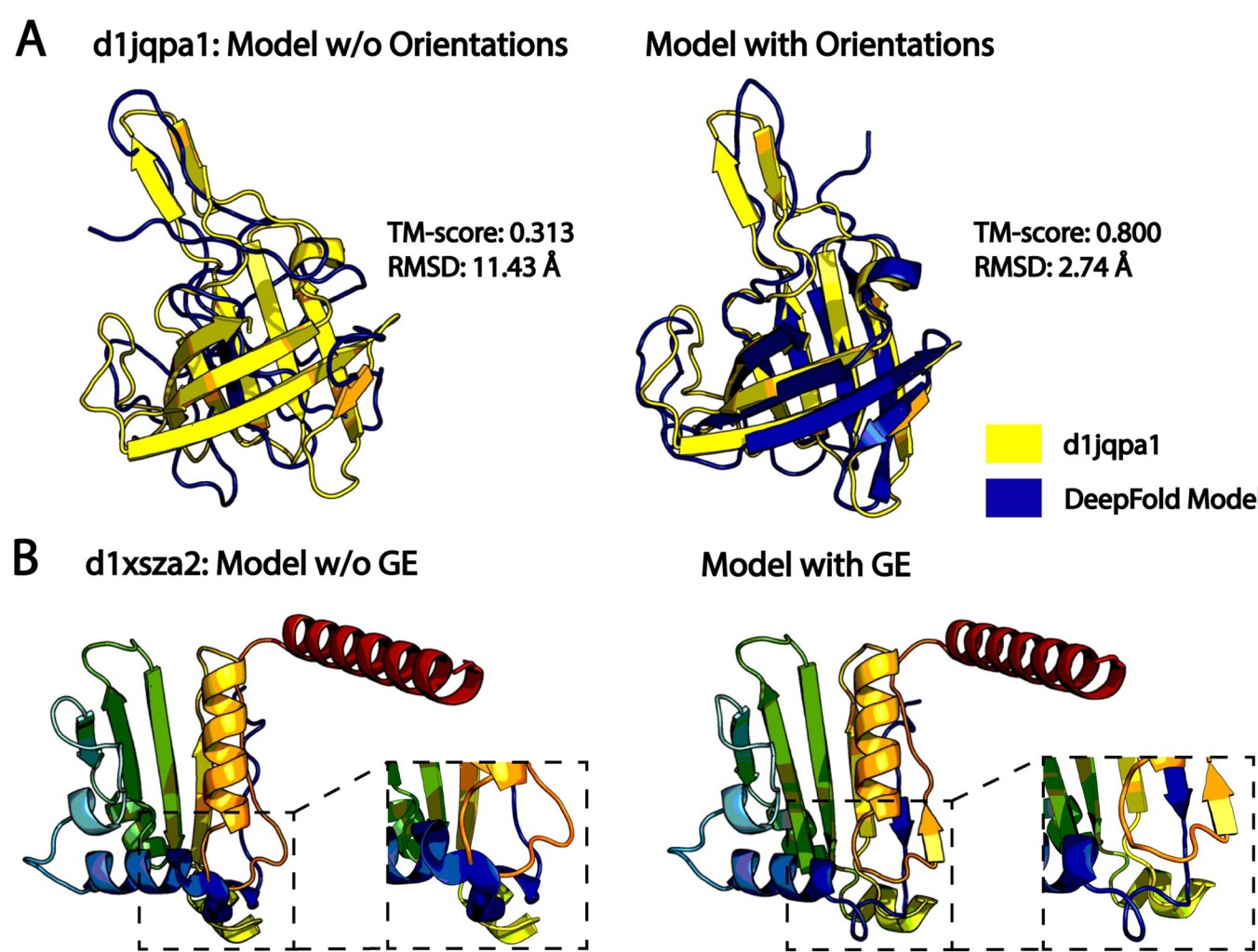

**Fig 3. Illustrative folding examples from DeepFold.** A) Case study from SCOPe protein d1jqpa1 that demonstrates the importance of inter-residue orientations for folding β-proteins, where the native structure is shown in yellow, and the superposed predicted models built without (left) and with (right) orientation restraints are shown in blue. B) Case study from SCOPe protein d1xsza2, which highlights the importance of the general energy function for improving the physical quality of the models. The models built without (left) and with (right) the general physical energy function are depicted in rainbow coloring, where the clashing region is shown in the inset on the left and the clashes have been resolved in the model built with the general energy function on the right.

saw that the major contributor to DeepFold's accuracy is the high number of accurately predicted restraints generated by DeepPotential, where their addition dramatically improved the average TM-score from 0.184 to 0.751 (Fig 2). Nevertheless, the physical energy function, which accounts for fundamental forces that drive protein folding, such as hydrogen bonding interactions and van der Waals clashes, plays an important role in improving the physical quality of the predicted models; this is especially true when the model quality is poor. As evidence, S4 Table lists several model quality metrics for models generated with and without the use of the GE function. On the overall test set of 221 hard protein targets, the inclusion of the GE potential provided a modest yet consistent enhancement in the physical model quality, as reflected in the improvement of the MolProbity score [21] from 1.735 to 1.692 with the addition of the GE function (S4 Table). Similar trends were observed for the secondary structure quality (SOV score [22]), the number of Ramachandran outliers, and the steric clash score

(S4 Table), all of which improved with the inclusion of the GE. The most notable improvement was observed in the clash score, which improved by 13.3% on the overall dataset.

More significant improvements were witnessed for the 16 targets with poor physical quality, as measured by a MolProbity score in the 50th percentile or lower from the PDB structures. For these targets, the physical energy function improved the average MolProbity score from 2.882 to 2.308, representing an improvement of 19.9% compared to 2.5% on the overall dataset. Similarly, these improvements were consistent across the SOV score, number of Ramachandran outliers, and the clash score for these targets. Again, the most dramatic improvement occurred for the clash score, which decreased from 17.5 to 8.6, representing an improvement of 50.9%. Fig 3B illustrates a case study from SCOPe protein d1xsza2, where models were generated with and without the inclusion of the general physical energy function. In the model built without the GE function, there are several residues that directly overlap each other leading to severe steric clashing, as shown in the inset. These clashes among other factors led to a model with a very high, and thus unfavorable, MolProbity score of 3.908 (3rd percentile) along with a very high clash score of 212.8. As shown in the inset in Fig 3B, these clashes were resolved with the inclusion of the GE potential and its term for van der Waals clashes, where the resulting model had a reduced MolProbity score of 1.624 (92nd percentile) and a low clash score of 1.2. Clearly, simply satisfying the geometric restraints provided by deep learning may lead to models that are physically unrealistic, where the introduction of physical energy terms may partially alleviate this problem.

## Comparison of DeepFold with other leading modeling methods

To further evaluate the performance of DeepFold, we compared the modeling results on the 221 test proteins with a leading contact map-based folding program (C-I-TASSER [23]), two top distance (DMPfold [24]) and distance/orientation-based (trRosetta [12]) methods, and the classic I-TASSER pipeline [7]. To provide a fair comparison, we used the same MSAs that DeepFold used, which were produced by DeepMSA2 [18] (see S1 Fig), for the deep learning restraint prediction by DMPfold, trRosetta and C-I-TASSER, as well as for template identification by LOMETS in I-TASSER and C-I-TASSER. Furthermore, templates with ≥30% sequence identity to the query were excluded from I-TASSER and C-I-TASSER.

As shown in Tables 1 and S5, the average/median TM-scores of the DeepFold models for the 221 test proteins were significantly higher than all the control methods. For instance, the average TM-score for the models produced by I-TASSER was only 0.383, where DeepFold achieved an average TM-score (0.751) that was 96.1% higher than I-TASSER with a *p*-value of 9.4E-80 as determined by a paired, two-sided Student's t-test (Table 1). This result is understandable as I-TASSER does not use any deep learning spatial restraints, making the modeling

**Table 1. Summary of the structure modeling results by DeepFold and the control methods on the 221 test proteins.** The *p*-values were calculated between DeepFold and the control methods using paired, two-sided Student's t-tests.

| Method | TM-score (*p*-value) | RMSD (*p*-value) | Correct Folds* | $TM_{DeepFold} > TM_{Method}$‡ |
|---|---|---|---|---|
| I-TASSER | 0.383 (9.4E-80) | 15.10 (7.1E-25) | 24.0% | 95.9% |
| C-I-TASSER | 0.584 (1.8E-55) | 8.89 (4.0E-26) | 67.0% | 95.9% |
| DMPfold | 0.657 (5.6E-37) | 7.81 (2.0E-18) | 79.6% | 92.3% |
| trRosetta | 0.694 (8.3E-24) | 6.81 (4.7E-09) | 85.5% | 87.8% |
| DeepFold | **0.751** | **5.61** | **92.3%** | - |

* This column represents the percent of proteins with TM-scores ≥0.5.

‡ This column indicates the percent of test proteins for which DeepFold generated a model with a higher TM-score than the control method.

accuracy more reliant on the templates, while, by design, all homologous templates were excluded for the Hard threading targets. The inclusion of deep learning contact maps into C-I-TASSER greatly increased the TM-score to 0.584. Nevertheless, DeepFold still achieved an average TM-score that was 28.6% higher than C-I-TASSER with a *p*-value of 1.8E-55. This is mainly due to the fact that DeepFold utilizes both distance and orientation restraints, which contain more detailed information than the contact maps used in C-I-TASSER [5]. The results for the median values were similar to the averages, where DeepFold achieved a median TM-score of 0.800, while I-TASSER and C-I-TASSER obtained median TM-scores of 0.357 and 0.607, respectively, which were significantly lower than DeepFold with p-values of 3.1E-37 and 1.9E-35 as determined by two-sided, non-parametric Wilcoxon signed-rank tests (S5 Table).

Interestingly, there were two targets (d1ltrd and d1nova) for which I-TASSER and C-I-TASSER produced models that were significantly more accurate than DeepFold. To examine the reason for the discrepancy in performance, S2 Fig depicts the models generated by I-TASSER, C-I-TASSER, and DeepFold superposed with the native structures along with the top templates used by I-TASSER and C-I-TASSER for these proteins. For d1ltrd, despite the fact that it was a hard threading target, LOMETS was able to identify a reliable template from the PDB (1prtI) with a coverage of 92.6% and a TM-score of 0.553; thus, both I-TASSER and C-I-TASSER constructed accurate models with TM-scores of 0.663 and 0.637, respectively. Conversely for DeepFold, the generated MSA contained few homologous sequences with a normalized number of effective sequences (or Neff, defined in S1 Text) of 0.42, resulting in inaccurate predicted restraints with an MAE of 2.60 Å for the top $2^*L$ distances. This ultimately lead DeepFold to produce a poor model with a TM-score of 0.326. Additionally, the contact precision for the top $L/2$ contacts used by C-I-TASSER was only 50.0%, which is largely why the C-I-TASSER model was worse than the I-TASSER model. Similarly, for d1nova, LOMETS was able to identify a reliable template (PDB ID 1hofC) with a coverage of 100% and a TM-score of 0.544, which resulted in accurate I-TASSER and C-I-TASSER models with TM-scores of 0.631 and 0.713 for the two methods, respectively. Again, for DeepFold, the generated MSA was shallow with a normalized Neff value of 9.40. Nevertheless, the predicted distance restraints were still accurate with an MAE of 0.90 Å for the top $2^*L$ distances; however, the predicted orientations were inaccurate, particularly the $\Omega$ orientation, which had an MAE of 31.3˚ for the top $2^*L$ restraints. This resulted in a model with a TM-score of 0.546, which still possessed a correct fold, but was worse than the models generated by I-TASSER and C-I-TASSER. Unlike the previous example, the C-I-TASSER model was closer to the native structure than the I-TASSER model for d1nova as the predicted contacts were accurate with a precision of 98.7% for the top $L/2$ contacts. These two examples highlight that even with the advances in deep learning methods, template-based modeling still remains important, particularly given the reliance of deep learning techniques on the generated MSAs, which may be lower quality than the identified templates for numerous targets.

DeepFold also outperformed two other leading distance (DMPfold) and distance/orientation-based (trRosetta) methods, where DMPfold achieved average/median TM-scores of 0.657/0.710 and trRosetta obtained average/median TM-scores of 0.694/0.749. Therefore, DeepFold's average/median TM-scores were 14.3%/12.7% higher than DMPfold and 8.2%/ 6.8% higher than trRosetta, where the differences were statistically significant with *p*-values of 5.6E-37/2.0E-34 and 8.3E-24/1.6E-26, respectively (see Tables 1 and S5). Furthermore, Fig 4 presents a head-to-head comparison of DeepFold with the control methods, where DeepFold outperformed trRosetta and DMPfold on 194 and 204 of the 221 test proteins, respectively. Compared to DMPfold, an obvious advantage of DeepFold is the use of inter-residue dihedral angle orientations, which resulted in a substantial TM-score increase for DeepFold as shown in Fig 2. Compared to trRosetta, since both methods use distance and orientation restraints,

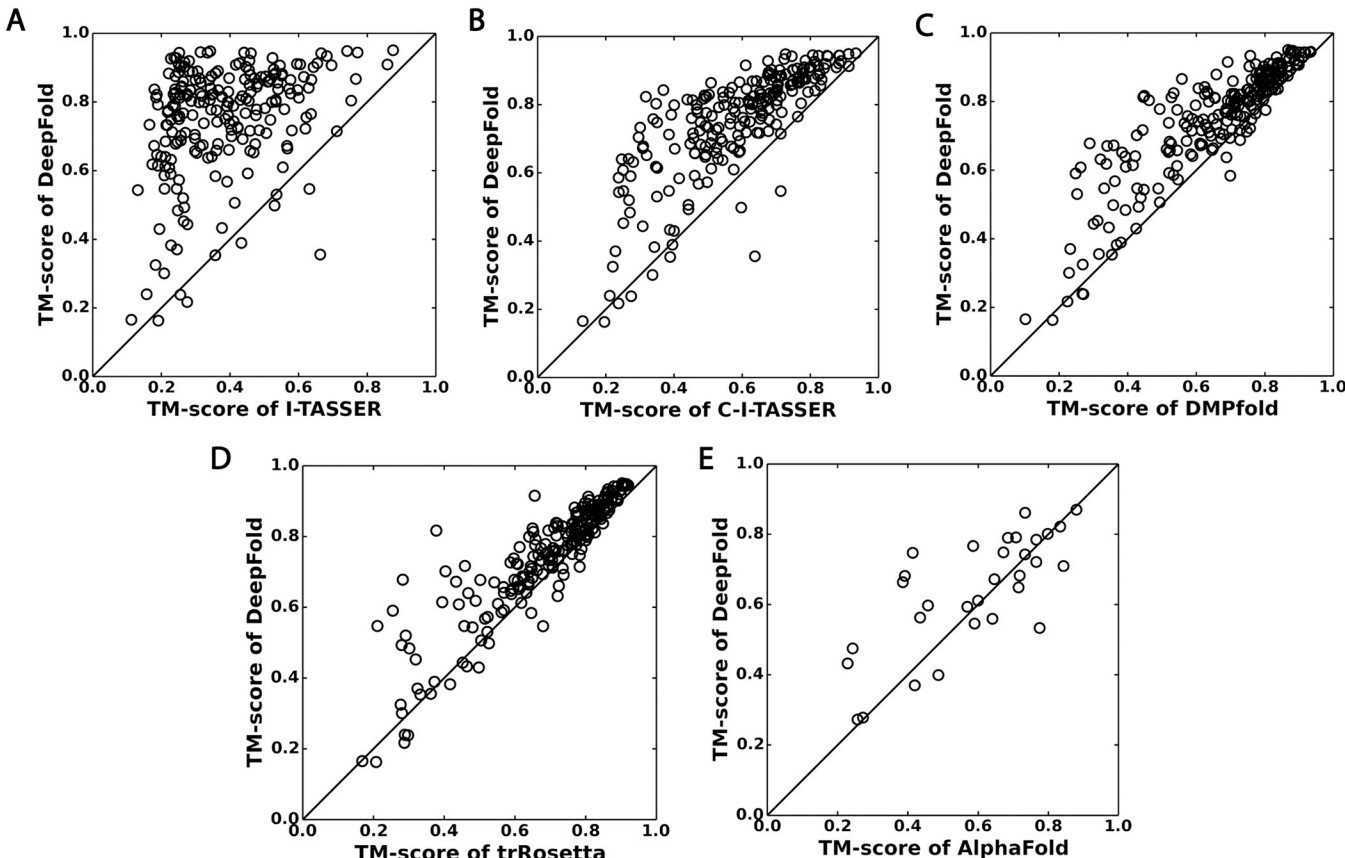

**Fig 4.** Head-to-head TM-score comparisons between DeepFold and other protein structure prediction methods: A) I-TASSER; B) C-I-TASSER; C) DMPfold; D) trRosetta; E) AlphaFold. (A-D) are based on the 221 Hard benchmark proteins, while (E) is on 31 FM targets from CASP13.

the major advantage of DeepFold is the high accuracy of the restraints generated by DeepPotential. Therefore, in S6 Table, we provide an accuracy comparison for the *Cβ* distance predictions by different programs, where the distance maps by DeepPotential had a significantly lower MAE to the native structures than those produced by both trRosetta and DMPfold across all cutoff values. In S7 Table, we also list the modeling results of trRosetta using the DeepPotential restraints. Although trRosetta+DeepPotential resulted in a higher average TM-score (0.735) than trRosetta alone, due to the use of the more accurate restraints from DeepPotential, the average TM-score of DeepFold was still significantly higher than that of trRosetta+-DeepPotential with a *p*-value of 3.9E-9. This is probably due to the unique DeepFold knowledge-based force field and the utilization of the additional *Cα* distance maps that are not used by trRosetta. In addition, the simultaneous optimization of the DeepFold force field with the L-BFGS search engine (see Methods) helped enhance the structure construction process.

Here, of particular interest is the modeling performance for those hard targets with very few effective sequences in their MSAs, which are the most difficult targets to fold using deep learning approaches. For this purpose, we collected a set of 16 targets with normalized Neff values less than 1.0 and calculated the TM-scores for the models produced by DeepFold, trRosetta, and DMPfold. On these targets, DeepFold achieved an average TM-score of 0.494, which was 40.3% higher than trRosetta (0.352) and 44.9% higher than DMPfold (0.341). In S3 Fig, we present a scatter plot of TM-score vs. the logarithm of the normalized MSA *Neff* value for the three methods on all 221 test proteins, where DeepFold demonstrated a lower correlation

between the TM-score and *Neff* value than trRosetta and DMPfold, which partially explains its superior performance.

It is of note that some of the proteins in the benchmark dataset may be homologous to the proteins that DeepFold and the other methods were trained on, as these deep learning methods often require a comprehensive set of training proteins to properly generalize. Thus, in S8 Table we depict the results for the selected methods on the 90 proteins in the benchmark dataset that shared <30% sequence identity to any of the training proteins used by DeepPotential. From the data in Tables S8 and 1, it can be seen that the performances of each of the methods, including DeepFold, were quite similar to the results on the overall benchmark dataset, where the accuracy of each of the deep learning methods on the 90 proteins was only slightly lower (~0.7–2.8% lower average TM-scores) than their accuracy on the 221 benchmark targets, which is largely due to the lower Neff values for the MSAs in the pruned dataset. Nevertheless, DeepFold still significantly outperformed each of the control methods on these targets.

Lastly, we compared the modeling accuracy of DeepFold with AlphaFold on the 31 CASP13 FM targets that the AlphaFold human group submitted models for (Fig 4E). Note, we could not benchmark the performance of AlphaFold on the 221 test proteins as the feature generation scripts and folding pipelines were not publicly available when this work was performed. It can be seen from Fig 4E that DeepFold outperformed AlphaFold on 20 of the 31 FM targets, where, on average, the TM-score of DeepFold was 0.636 compared to 0.589 for AlphaFold (*p*-value = 0.025, S9 Table). It is also important to note that the AlphaFold human group performed thousands of different optimization runs for the CASP13 targets as reported [11], while DeepFold only used a single optimization run in this study.

## Comparison of DeepFold with the most recently developed methods: AlphaFold2 and RosettaFold

Since DeepFold uses restraints from DeepPotential, which was developed before the advances made by AlphaFold2 [25] in CASP14, it is also of interest to compare the results against the most recent self-attention-based neural network methods, namely, AlphaFold2 and Rosetta-Fold [26]. Thus, in S4A–S4C Fig, we provide a head-to-head comparison of the DeepFold modeling results utilizing the restraints from DeepPotential with RosettaFold and AlphaFold2 on the 221 test proteins in terms of the model TM-scores, where the results are summarized in S10 Table. Overall, the average TM-score of the RosettaFold end-to-end pipeline was 0.812 and the average TM-score of the Pyrosetta version was 0.838, which were higher than the results by DeepFold (TM-score = 0.751) with *p*-values of 3.6E-10 and 8.0E-22, respectively. Similarly, the average TM-score of AlphaFold2 was 0.903, which was higher than DeepFold with a p-value of 1.4E-49. These results were expected given that the advances in deep self-attention neural networks and end-to-end training by AlphaFold2 and, subsequently, Rosetta-Fold showed greatly improved modeling accuracy over previously introduced convolutional ResNet architectures, such as DeepPotential.

Notably, there were 7 targets for which DeepFold outperformed AlphaFold2. In S5 Fig, we illustrate two examples where DeepFold generated models that were significantly more accurate than AlphaFold2. The first example is from SCOPe protein d1a34a, for which DeepFold generated a model with a TM-score of 0.613, while AlphaFold2 generated a model with a TM-score of 0.242. For this target, DeepMSA2 was not able to identify any sequence homologs, resulting in an MSA composed of only the query sequence and an extremely low normalized Neff value of 0.08. Nevertheless, DeepPotential generated accurate restraints with an MAE of 1.10 Å for the top $2^*L$ distances, resulting in a higher quality model than that produced by AlphaFold2. The second example is from SCOPe protein d1s2xa, for which DeepFold

generated a model with a TM-score of 0.590, while AlphaFold2 generated a model with a TM-score of 0.369. Again, for this target, DeepMSA2 was only able to identify two sequence homologs, which resulted in a very low normalized Neff value of 0.15. Additionally, the DeepPotential restraints were fairly inaccurate with an MAE of 2.54 Å for the top $2^*L$ distances and 59.29° for the $2^*L$ Ω orientations. Interestingly, even though the orientation restraints were inaccurate, their inclusion greatly improved the modeling accuracy, as the model built using only the contact and distance restraints possessed a low TM-score of 0.268, while the model built using the full set of contact/distance and orientation restraints had a TM-score of 0.514. Moreover, the addition of the general knowledge-based energy function further improved the TM-score to 0.590. This suggests that even when inaccurate, the combination of various restraints with a general energy function may act synergistically to filter out inaccuracies in the predictions. It is also noteworthy that the two preceding examples were from proteins with few to no homologous sequences. In fact, if we consider the 5 proteins in the benchmark dataset with the least homologous sequence information (<3 sequence homologs) and normalized Neff values <0.20, DeepFold generated more accurate models than AlphaFold2 for 4 of these targets, where the average TM-score of DeepFold was 0.528 compared to 0.398 for AlphaFold2. This suggests that, while deep self-attention-based protein structure prediction approaches have demonstrated an improved ability to fold proteins with few sequence homologs, the performance on the most extreme cases with few to no sequence homologs remains to be improved.

Lastly, given the importance of the most recent advances in protein structure prediction, we sought to determine whether or not they could be incorporated into DeepFold to further improve its performance. To answer this question, we utilized the restraints taken from RosettaFold, including the Cβ distances and orientations, as well as the Cα distances/contacts and Cβ contacts from DeepPotential to guide the DeepFold simulations. The results of this analysis are depicted in S11 Table and S4D–S4F Fig, which present head-to-head comparisons between DeepFold utilizing the combined restraints with RosettaFold and AlphaFold2 in terms of the model TM-scores on the 221 benchmark proteins. The results show that with the combined RosettaFold and DeepPotential restraints, DeepFold achieved an average TM-score of 0.844, which was higher than that attained by the end-to-end (TM-score = 0.812) and Pyrosetta (TM-score = 0.838) versions of RosettaFold with *p*-values of 2.4E-11 and 1.2E-2, respectively. These data demonstrate that the DeepFold knowledge-based force field and DeepPotential contact and Cα distance restraints may improve the results obtained by RosettaFold. Additionally, they show that DeepFold is a versatile platform that can be easily adapted for any future advances in state-of-the-art deep learning restraint predictors.

## DeepFold greatly improves the accuracy and speed of protein folding over classical *ab initio* methods

Rosetta [14] and QUARK [8] are two of the most well-known fragment-assembly methods and have been consistently ranked as the top methods for *ab initio* protein structure prediction in previous CASP experiments [3, 27, 28]. However, a major drawback of the traditional *ab initio* folding approaches is that their modeling performance drops as the protein length increases, making them significantly less reliable for modeling larger protein structures composed of more than 150 residues [1]. To examine the impact of deep learning on *ab initio* structure prediction for long protein sequences, we compared DeepFold to both Rosetta and QUARK, where Fig 5C depicts the TM-scores of DeepFold, QUARK, and Rosetta vs protein length. The data show that the performance of DeepFold remained consistent as the protein length increased, where the average TM-score for large proteins composed of 350–450 residues

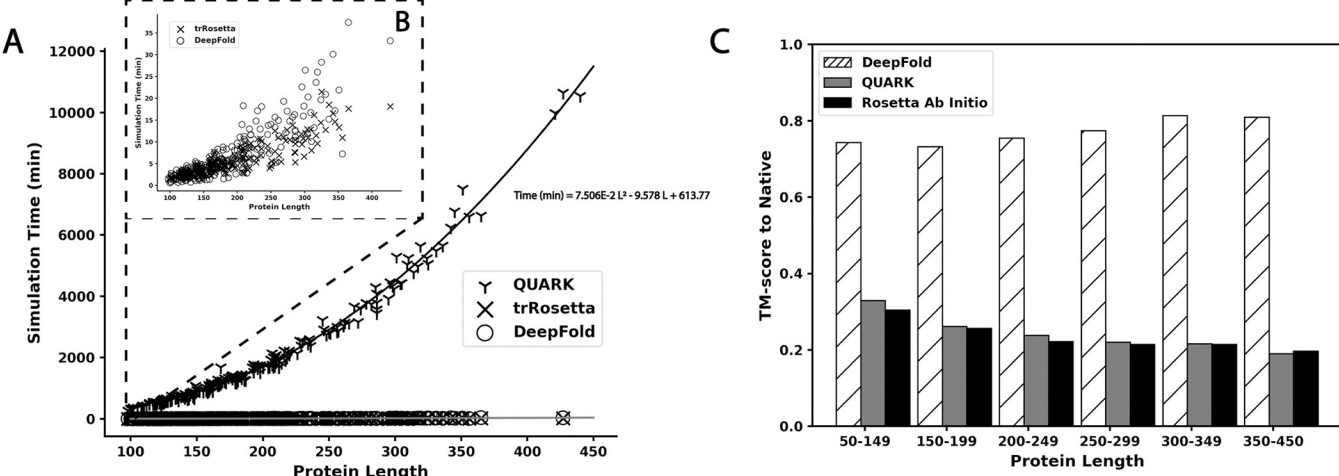

**Fig 5. Dependence of the simulation time and TM-score on protein length.** A) Simulation runtime for QUARK, trRosetta, and DeepFold in minutes plotted against the protein length. B) A close up of the runtime vs protein length for DeepFold and trRosetta. C) Analysis of the average TM-score for DeepFold, QUARK, and Rosetta across different protein length ranges.

was in fact higher than that for the small proteins in the test set with lengths <150 residues (0.809 vs. 0.742), mostly due to the more favorable MSAs collected for the set of larger proteins. However, the performance of both QUARK and Rosetta noticeably decreased as the protein length increased; the average TM-score for proteins with lengths less than 150 residues was 0.329 for QUARK and 0.304 for Rosetta but was only 0.190 and 0.196 for QUARK and Rosetta, respectively, on proteins with lengths between 350 and 450 residues. From these results, DeepFold outperformed QUARK and Rosetta remarkably on the overall dataset and especially on the longest proteins in the dataset, for which the average TM-score of DeepFold was 325.8% higher than QUARK and 312.8% higher than Rosetta.

Another major limitation of fragment-assembly approaches is that they require lengthy simulations to adequately explore the immense structure space available. In Fig 5A and 5B, we list a comparison of the folding simulation time requirement for DeepFold and the QUARK fragment assembly approach for different protein lengths. The results show that the speed of DeepFold is orders of magnitude faster than QUARK, especially for large proteins. Note that we ran QUARK using 5 separate trajectories in parallel and the run times shown in Fig 5A are the average run time across all 5 simulation trajectories. Thus, if the simulations were run sequentially, the run time would be 5 times longer, which further accentuates the cost of fragment assembly. Therefore, while fragment assembly requires hours to days to fold a protein, DeepFold requires only seconds to minutes. Overall, the average run time of DeepFold on the test set was 6.98 minutes, while the average for QUARK was 1830.82 minutes for an average protein length of 188.1 residues. This indicates that QUARK requires 262.3 times the computing time that DeepFold requires for one simulation trajectory, and the difference was even greater as the sequence length increased. Overall, the run time of DeepFold was similar to trRosetta, which required 5.48 minutes to construct models on the test dataset on average. Of particular importance is that the greatly reduced folding times did not cause the model quality to deteriorate for larger proteins, demonstrating the ability of deep learning restraints to effectively smooth the energy landscape, thereby allowing rapid and accurate optimization across protein lengths.

## Gradient-based protein folding requires a high number of deep learning restraints

The success of rapid L-BFGS-based protein folding approaches raises the question on what the role of fragment assembly is in protein structure prediction. As L-BFGS and other gradient-based methods are essentially local optimization techniques that may be prone to becoming trapped in local energy minima, the more extensive conformational sampling performed by fragment assembly may still be necessary in the absence of a high number of deep learning spatial restraints.

To examine this hypothesis, Fig 6A depicts the TM-score for L-BFGS-based protein folding simulations using different numbers of spatial restraints. Consistent with the data in Fig 2, Fig 6A shows that only using the GE function to guide the L-BFGS simulations resulted in a poor average TM-score of 0.184, which was significantly lower than that obtained by QUARK (TM-score = 0.274), which uses a knowledge-based energy function without deep learning restraints [8]. This indicates the frustration of the baseline energy force field of DeepFold, which cannot be quickly explored with gradient-based methods. Inclusion of the top $L$ all-range Cβ distances slightly improved the TM-score to 0.186, and at least the top $5^*L$ distances were required to improve the TM-score to a significant degree. In order to achieve a performance that was better than QUARK, the L-BFGS simulations required $10^*L$ Cβ distance restraints, where the average TM-score using this number of restraints was 0.323. The inclusion of more distance restraints, such as the top $15^*L$ and $20^*L$ restraints, steadily improved the average TM-score to 0.392 and 0.453, respectively.

However, our tests showed that setting a specific probability cutoff for the selection of distance restraints allowed the method to achieve the best result. In DeepFold, all distances with a

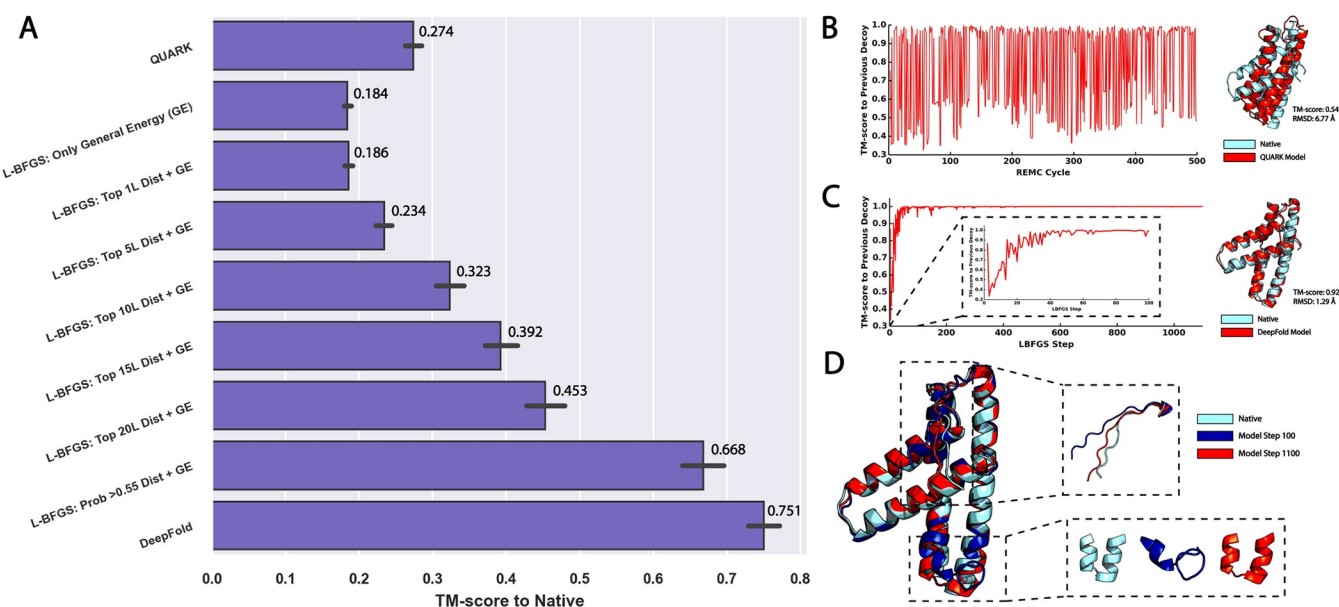

**Fig 6. Comparison of DeepFold and QUARK modeling results.** A) Evaluation of the modeling accuracy of QUARK and DeepFold guided by different numbers of spatial restraints, where the top $n^*L$ distances were selected by sorting the Cβ distances according to their predicted probabilities. B) Analysis of the conformational changes that occured during the QUARK fragment assembly simulations. The figure plots the TM-score of the decoy at REMC cycle $i$ compared to the decoy at the previous cycle $i-1$. The right hand side shows the final QUARK model in red superposed with the native structure in cyan. C) Analysis of the conformational changes that occured during the DeepFold simulations. The figure plots the TM-score of the decoy at L-BFGS step $i$ compared to the decoy at the previous step $i-1$, where the right hand side shows the final DeepFold model in red superposed with the native structure in cyan. D) Comparison between the DeepFold model at L-BFGS step 100 (blue) with the model at step 1100 (red) and the experimental structure (cyan). The insets show the areas of the structure that changed the most after the 100th L-BFGS step.

probability $>0.55$ were selected for inclusion in the L-BFGS optimization procedure, which corresponded to an average of ~$93^*L$ distance restraints on the test set, increasing the TM-score to 0.668. Overall, the addition of the full set of DeepPotential restraints (including contacts, C$\alpha$ distance and orientations in addition to the C$\beta$ distances) increased the accuracy by an additional 12.4%, resulting in a TM-score of 0.751 for the full pipeline. Thus, it is clear that L-BFGS requires a high number of spatial restraints in order to adequately smooth the energy landscape and make gradient-based protein folding feasible.

## Case study reveals drastically different dynamics in Monte Carlo and L-BFGS folding simulations

To further illustrate the differences in the sampling procedures for the fragment assembly method, QUARK, and the L-BFGS optimization approach, DeepFold, we present in Fig 6B–6D a case study from the amino terminal domain of enzyme I from E. coli (SCOPe ID: d1zyma1). Both DeepFold and QUARK generated a correct fold for this target, where the TM-score of the model produced by QUARK was 0.547 and the TM-score for the DeepFold model was very high at 0.923 with an RMSD of 1.29 Å, indicating a close atomic match to the experimental structure.

To show the conformational changes during the QUARK folding simulations, Fig 6B depicts the TM-score of the conformation for the last replica at REMC cycle *i* relative to the conformation of the previous decoy at cycle *i*-1. From the figure, it can be seen that large changes in the conformation occur throughout the simulation due to the global conformational searching and replica exchange steps. On the other hand, the opposite trend was observed for the L-BFGS folding simulations shown in Fig 6C, during which large conformational changes occurred early on in the simulation, and the global fold of the protein was largely determined by the 100[th] L-BFGS step. After that, only small fluctuations in the conformation occurred, where the L-BFGS optimization quickly converged and did not extensively sample the structure space due to the nature of the local optimization of the smooth energy landscape produced by the large number of deep learning restraints.

Moreover, Fig 6D depicts the DeepFold models at L-BFGS steps 100 and 1100 superposed with the experimental structure. While the global fold of the model was determined by the 100[th] L-BFGS step, substantial conformational changes occurred during the later L-BFGS steps at the two regions, namely the highlighted terminal coil and core helix regions, which were poorly formed at step 100 due to the inconsistency in the spatial restraints in these sections. For the helix region in particular, the model at step 100 had poorly formed secondary structure as well as severely clashing segments. These errors were gradually corrected over the remaining 1000 L-BFGS steps. Therefore, while the global folds of proteins may quickly be determined by the consensus DeepPotential restraints during the L-BFGS simulations, additional steps are often needed to precisely fine-tune the model quality under the guidance of the atomic force field.

## Conclusions

We developed an open-source program (DeepFold) to quickly construct accurate protein structure models from deep learning-based potentials. DeepFold significantly outperformed other *ab initio* structure prediction methods such as Rosetta, QUARK, I-TASSER, C-I-TASSER, DMPfold, and trRosetta on the test set of 221 Hard threading targets, and AlphaFold on the CASP13 FM targets. The impact of deep learning on DeepFold was best highlighted by the benchmark test with Rosetta, QUARK and I-TASSER, which represent the top traditional FM and TBM methods. On the benchmark dataset, Rosetta, QUARK and I-TASSER were only

able to generate correctly folded models for 0.9%, 2.7% and 24.0% of the proteins, respectively, while DeepFold successfully folded 92.3% of the test proteins with an average TM-score of 0.751, compared to 0.260, 0.274, and 0.383 for Rosetta, QUARK and I-TASSER, respectively.

Furthermore, the average TM-score of DeepFold was 7.8% and 13.9% higher than the other leading deep learning-based methods, DMPfold and trRosetta, respectively, starting from the same MSAs. It was also 8.0% higher than AlphaFold on the 31 CASP13 FM targets. Of particular interest is the performance on the hardest targets in the dataset with very shallow MSAs (i.e., with normalized Neff values less than 1.0), where the average TM-score of DeepFold was 40.3% higher than trRosetta and 44.9% higher than DMPfold. On top of the improved accuracy, DeepFold had a similar running time as other gradient descent-based approaches such as trRosetta, but was more than 200 times faster than the traditional fragment-assembly-based approaches. The success of DeepFold is mainly due to the effective combination of the inherent knowledge-based potential with the high number of accurately predicted spatial restraints that help smooth the energy landscape, making L-BFGS optimization tractable.

Despite the success, significant improvements may still be made. For example, the use of attention-based networks [25, 29, 30], especially an end-to-end learning protocol [25], should help further improve the prediction accuracy of DeepFold. Given that the main input features to DeepPotential are derived from co-evolutionary analyses, DeepFold often requires that the input MSAs contain a sufficient number of effective sequences to enable determination of the co-evolutionary relationships between protein residues. Despite the fact that the quality of the DeepFold models was considerably less dependent on the MSA quality than other methods such as DMPfold and trRosetta, the use of a transformer architecture should help further enhance the performance of DeepPotential for those targets with poor MSA quality and few homologous sequences by self-attention based, iterative MSA refinement. This can be illustrated by the comparison of DeepFold with the most recent methods, RosettaFold and Alpha-Fold2, which achieved higher TM-scores on the benchmark targets. Nevertheless, when utilizing the combined RosettaFold and DeepPotential restraints, DeepFold was able to outperform both the end-to-end and distance-based versions of RosettaFold, demonstrating that it is a versatile platform that can be easily adapted for advances in the state of the art. Meanwhile, DeepFold outperformed AlphaFold2 on 4 out of the 5 targets with the least homologous sequence information (normalized Neff <0.2), revealing that there is significant room for improvement on very difficult modeling targets.

Furthermore, more efficient and precise MSA construction strategies should be developed to improve the MSA quality and reduce the time required to search the various sequence databases. The need to increase the searching efficiency is particularly important as the increase in the size of the sequence databases, mainly the metagenomics databases, is a double-edged sword. While it enables the collection of more sequences, it also greatly increases the time and computational resources necessary to search the sequence databases and the potential for false negative sequence samples due to the increase in noise. For example, searching a 150-residue protein through MetaClust, which is approximately 100 GB, using DeepMSA2 requires around 1 hour with 1 CPU; however, searching the same protein through the 5TB JGI metagenome database is dramatically more expensive, requiring approximately 4 hours using 50 CPUs. This issue is particularly important for hard modeling targets, which often require extensive homologous sequence detection. As evidence of this, in S6 Fig, we plot the number of times each of the 7 MSAs produced by DeepMSA2 were selected for the 221 benchmark targets. From the figure, it can be seen that ~55% of the targets required searching beyond the MetaClust database, while only ~15% did not require searching through any metagenomics database. Meanwhile, incorrectly collected MSAs, despite having a high number of homologous sequences, can negatively impact the modeling results as witnessed in the CASP experiments [31]. The

use of a targeted MSA generation protocol that focuses on searching sequences related to the target protein's biome represents a promising strategy for improving the speed and quality of the MSA generation and the accuracy of the final 3D structure modeling [32].

## Methods

DeepFold is an algorithm that can quickly construct accurate full-length protein structure models from deep learning restraints and consists of three main steps: MSA generation by DeepMSA2, spatial restraint prediction by DeepPotential, and L-BFGS folding simulations, as depicted in Fig 1.

### MSA generation by DeepMSA2

DeepMSA2 is an extension of DeepMSA [33] for iterative MSA collection, where the new components include an additional pipeline to search larger sequence databases and a novel MSA selection method based on predicted contact maps (see S1 Fig). Briefly, DeepMSA2 collects 7 candidate MSAs by iteratively searching whole-genome (Uniclust30 and UniRef90) and metagenome (Metaclust, BFD, and Mgnify) sequence databases. The first 3 MSAs are generated using the same procedure as DeepMSA (i.e., dMSA in S1 Fig), where the query sequence is first searched through Uniclust30 (2017_04) by HHblits2 to create MSA-1. Next, the sequences identified by Jackhmmer and HMMsearch are used to construct a custom HHblits database, against which HHblits2 is run starting from the MSA generated in the previous stage to generate MSA-2 and MSA-3, respectively. The four remaining MSAs are generated using a procedure called quadruple MSA (qMSA in S1 Fig), which uses HHblits2 to search the original query sequence against the Uniclust30 database (version 2020_01) to create MSA-4. Next, the sequences detected by Jackhmmer, HHblits3, and HMMsearch through the UniRef90, BFD, and Mgnify databases are used to construct custom HHblits-style databases, against which HHblits2 is employed to search starting from the MSAs generated by the previous stages to create MSA-5, MSA-6, and MSA-7, respectively. To select the final MSA, a quick TripletRes contact map prediction [34] is run starting from each of the 7 MSAs, where the MSA with the highest cumulative probability for the top $10^*L$ all-range contacts is selected as the final MSA.

### Spatial restraint prediction by Deep Potential

Starting from the selected MSAs, two sets of 1D and 2D features are extracted. The 2D features include the raw coupling parameters from the pseudo likelihood maximized (PLM) 22-state Potts model and the raw mutual information (MI) matrix, where the 22 states of the Potts model represent the 20 standard amino acids, a non-standard amino acid type, and a gap state. Here, a Potts model is a specific type of Markov Random Field (MRF) model that is widely used in protein structure prediction [35–38]. Briefly, an MRF is a graphical model that represents each column of an MSA as a node that describes the distribution of amino acids at a given position (Potts model field parameters), where the edges between nodes indicate the joint distributions of amino acids at each pair of positions. The 2D coupling parameters can then be determined from the edge weights, where residue pairs that exhibit correlated mutation patterns will possess greater edge weights, which can be used to infer positions that should be closer together in 3D space. This is based off of the intuition that if two residues are in contact with each other, then when one residue mutates, the contacting residue should also mutate in order to preserve the interaction. In DeepPotential, CCMpred [38] is used to fit the Potts model. The corresponding parameters for each residue pair in the PLM and MI matrices are extracted as additional features that measure query-specific co-evolutionary information in an

MSA. The 1D features contain the Potts model field parameters, Hidden Markov Model (HMM) features, and the self-mutual information, along with the one-hot representation of the MSA and other descriptors, such as the number of sequences in the MSA.

Next, these 1D and 2D features are fed into deep convolutional residual neural networks separately, where each of them is passed through a set of one-dimensional and two-dimensional residual blocks, respectively, and are subsequently tiled together. The tiled feature representations are considered as the input of another fully residual neural network which outputs the inter-residue interaction terms, including Cα-Cα distances, Cβ-Cβ distances, and the inter-residue orientations (Fig 1). Here, the predicted spatial restraints are represented using various bins that correspond to specific distance/angle values, where DeepPotential predicts the probability that the spatial restraints fall within the specific bins. For example, for the Cα and Cβ distances, the predictions are divided into 38 bins, where the first bin represents the probability that the distance is <2Å and the final bin represents the probability that the distance is ≥20Å. The remaining 36 bins represent the probability that the distance falls in the range [2Å, 20Å], where each bin has a width of 0.5 Å. On the other hand, the 3 orientation features, as defined in S7 Fig, are predicted using a bin width of 15˚ with an additional bin to indicate whether there is no interaction between the two residues (i.e., Cβ-Cβ distance ≥20Å). The DeepPotential models were trained on a set of 26,151 non-redundant proteins collected from the PDB at a pair-wise sequence identity cutoff of 35%.

## DeepFold Force Field

The DeepFold energy function is a linear combination of the following terms:

$$E_{DeepFold} = (E_{C\beta dist} + E_{C\alpha dist} + E_{C\beta cont} + E_{C\alpha cont} + E_{\Omega} + E_{\theta} + E_{\varphi}) + (E_{hb} + E_{vdw} + E_{tor}) \qquad (1)$$

where the first seven terms $E_{C\beta dist}$, $E_{C\alpha dist}$, $E_{C\beta cont}$, $E_{C\alpha cont}$, $E_{\Omega}$, $E_{\theta}$, and $E_{\varphi}$ account for the predicted Cβ–Cβ distances, Cα–Cα distances, Cβ–Cβ contacts, Cα–Cα contacts, and three inter-residue orientation angles by DeepPotential; and the last three terms $E_{hb}$, $E_{vdw}$, and $E_{tor}$ denote the generic energy terms for hydrogen bonding, van der Waals clashes, and backbone torsion angles, respectively.

Overall, the DeepFold force field consists of 24 weighting parameters, where the weights given to each of the deep learning restraints were separated into short ($1 < |i-j| \leq 11$), medium ($11 < |i-j| \leq 23$) and long-range ($|i-j| > 23$) weights, which were determined by maximizing the TM-score on the training set of 257 non-redundant, Hard threading targets collected from the PDB that shared <30% sequence identity to the test proteins. Briefly, all the weights were initialized to 0, then the weight for each individual energy term was varied one-at-a-time by an increment of 0.25 in the range from [0, 25] and the DeepFold folding simulations were run using the new weights. The weight for each term that resulted in the highest average TM-score on the training set was accepted. After the initial weighting parameters were determined, 3 more optimization runs were carried out, where the weight for each energy term was again varied in a range from [0, 25] using an increment of 0.1 and the weighting parameters that resulted in the highest average TM-score on the training set were accepted. A final optimization run was carried out, where the weights were perturbed by [–2, 2] from their previously accepted values using an increment of 0.02 to precisely fine-tune their values. The details of each energy term are further explained in S2 Text in the SI. Since DeepPotential provides the bin-wise histogram probability of the spatial descriptors, these terms are further fit with cubic spline interpolation to facilitate the implementation of the L-BFGS optimization, which requires a continuously differentiable energy function.

## L-BFGS Folding Simulations

A protein structure in DeepFold is specified by its backbone atoms (N, H, Cα, C, and O), Cβ atoms and the side-chain centers of mass (S8 Fig). The initial conformations are generated from the backbone torsion angles ($\phi$, $\psi$) predicted by ANGLOR through a small, fully-connected neural network [39], where the cartesian coordinates of the backbone atoms are determined using simple geometric relationships, assuming ideal bond length and angle values. The conformational search simulations are performed using L-BFGS, with bond lengths and bond angles fixed at their ideal values, and the optimization is carried out on the backbone torsion angles.

Here, L-BFGS is a gradient-descent based optimization method that is a limited memory variant of the Broyden-Fletcher-Goldfarb-Shanno (BFGS) algorithm. At each step $k$, the search direction $d_k$ of the simulation is calculated by

$$d_k = -H_k^{-1} \cdot \nabla E_{DeepFold}(x) \tag{2}$$

where $H_k^{-1}$ is an estimate for the inverse Hessian matrix and $\nabla E_{DeepFold}(x)$ represents the gradient of $E_{DeepFold}(x)$ with respect to the backbone torsion angles $x = (\phi, \psi)$. The value of $H_k^{-1}$ at step $k = 0$ is set to the identity matrix, $I$, and the value of $H_{k+1}^{-1}$ is obtained following the BFGS formulation

$$\begin{cases} H_{k+1}^{-1} = V_k^T H_k^{-1} V_k + \rho_k s_k s_k^T \\ V_k = I - \rho_k y_k s_k^T \\ \qquad \rho_k = 1/\ y_k^T s_k \end{cases} \tag{3}$$

where $s_k = x_{k+1} - x_k$ and $y_k = \nabla E_{DeepFold}(x_{k+1}) - \nabla E_{DeepFold}(x_k)$. $H_{k+1}^{-1}$ can be computed recursively by storing the previously calculated values of $s_k$ and $y_k$. To preserve memory, L-BFGS only stores the last $m$ values of $s_k$ and $y_k$. Thus, $H_{k+1}^{-1}$ is calculated by

$$H_{k+1}^{-1} = \left(\prod_{i=k}^{k-\hat{m}+1} V_i^T\right) H_0^{-1} \left(\prod_{i=k-\hat{m}+1}^{k} V_i\right) + \sum_{j=k}^{k-\hat{m}+1} \left(\prod_{i=k+1}^{j+1} V_i\right) \rho_k s_k s_k^T \left(\prod_{i=j+1}^{k} V_i\right) \tag{4}$$

where $\hat{m} = min(k, m-1)$ and $m$ is set to 256 in DeepFold. Once the search direction $d_k$ is decided, the torsion angles for the next step are updated according to

$$\begin{cases} \phi_{k+1} = \phi_k + \alpha_k d_k \\ \psi_{k+1} = \psi_k + \alpha_k d_k \end{cases} \tag{5}$$

The value of $\alpha_k$ is determined using the Armijo line search technique [40] and dictates the extent to move along the given search direction. In DeepFold, a maximum of 10 L-BFGS iterations are performed with 2,000 steps each, or until the simulations converge. The final model is selected as the one with the lowest energy produced during the folding simulations.

## Supporting information

**S1 Table. Impact of the different components of the DeepFold energy function on the structure modeling accuracy.**
(PDF)

**S2 Table. Mean absolute error (MAE) between the top specified number of long-range distance restraints predicted by DeepPotential and the models built without (GE+Cont+Dist) and with (GE+Cont+Dist+Orien) inter-residue orientations.**
(PDF)

**S3 Table. DeepFold results on the 38 β-proteins in the test set with and without orientation restraints.**
(PDF)

**S4 Table. Impact of the general statistical energy function on DeepFold's modeling performance.**
(PDF)

**S5 Table. Non-parametric analysis for DeepFold and the control methods on the 221 test proteins.**
(PDF)

**S6 Table. Top long-range distance MAE by different distance predictors on the 221 test proteins.**
(PDF)

**S7 Table. Modeling results for trRosetta using DeepPotential's spatial restraints vs Deep-Fold.**
(PDF)

**S8 Table. Modeling results for DeepFold and the control methods on the 90 test proteins that were non-redundant to the training set of DeepPotential.**
(PDF)

**S9 Table. Modeling results for DeepFold and AlphaFold on the 31 CASP13 targets.**
(PDF)

**S10 Table. Modeling results of DeepFold using the DeepPotential restraints vs Rosetta-Fold/AlphaFold2 on the 221 test proteins.**
(PDF)

**S11 Table. Modeling results of DeepFold using the combined RosettaFold/DeepPotential restraints vs RosettaFold/AlphaFold2 on the 221 test proteins.**
(PDF)

**S12 Table. Selection of the first well width ($d_b$) in the contact potential for various protein lengths ($L$).**
(PDF)

**S1 Fig.** DeepMSA2 pipeline, which contains three approaches, (A) dMSA, (B) qMSA, and (C) MSA selection.
(PDF)

**S2 Fig. Case study from two proteins for which I-TASSER/C-I-TASSER significantly outperformed DeepFold.**
(PDF)

**S3 Fig. Model TM-score vs. the logarithm of the MSA Neff value for DeepFold, trRosetta, and DMPfold.**
(PDF)

**S4 Fig. Head-to-head comparison between DeepFold and RosettaFold/AlphaFold2 on the 221 Hard benchmark targets.**
(PDF)

**S5 Fig. Case study from two proteins for which DeepFold significantly outperformed AlphaFold2.**
(PDF)

**S6 Fig. Histogram distribution of the number of times each of the 7 DeepMSA2 MSAs were selected for the 221 Hard benchmark targets.**
(PDF)

**S7 Fig. Definition of the inter-residue orientations predicted by DeepPotential.**
(PDF)

**S8 Fig. Depiction of the reduced model used to represent protein conformations during the DeepFold folding simulations.**
(PDF)

**S1 Text. MSA Neff value calculation.**
(PDF)

**S2 Text. Description of the DeepFold energy function.**
(PDF)

## Acknowledgments

We thank Dr. Wei Zheng for a portion of the design of Fig 1 and S1 Fig.

## Author Contributions

**Conceptualization:** Yang Zhang.

**Data curation:** Robin Pearce.

**Funding acquisition:** Yang Zhang.

**Investigation:** Robin Pearce.

**Methodology:** Robin Pearce, Yang Li.

**Software:** Robin Pearce, Yang Li.

**Supervision:** Gilbert S. Omenn, Yang Zhang.

**Writing – original draft:** Robin Pearce, Yang Zhang.

**Writing – review & editing:** Yang Li, Gilbert S. Omenn.

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
