## [Decision Letter · Decision Letter 0]

10 Mar 2022

Dear Dr. Zhang,

Thank you very much for submitting your manuscript "Fast and Accurate Ab Initio Protein Structure Prediction Using Deep Learning Potentials" for consideration at PLOS Computational Biology.

As with all papers reviewed by the journal, your manuscript was reviewed by members of the editorial board and by several independent reviewers. In light of the reviews (below this email), we would like to invite the resubmission of a significantly-revised version that takes into account the reviewers' comments.

We cannot make any decision about publication until we have seen the revised manuscript and your response to the reviewers' comments. Your revised manuscript is also likely to be sent to reviewers for further evaluation.

Sincerely,

Christos A. Ouzounis

Associate Editor

PLOS Computational Biology

Arne Elofsson

Deputy Editor

PLOS Computational Biology

Reviewer's Responses to Questions

**Comments to the Authors:**

Reviewer #1: The authors describe in this paper the important progress that they have made in protein structure predictions with their new DeepFold pipeline by including the predicted contacts, a deep learning-based potential and a knowledge-based statistical force field. Despite the huge publicity for Alphafold2, predicting protein structres remains an important research problem. One only has to inspect the Alphafold database to see the large number of extended segments being specified as being of low confidence.

Second paragraph – ‘use of deep learning techniques to predict spatial restraints’ is misleading since it is primarily the data from sequence correlations that lead to these gains, rather than the machine learning itself. There are new gains that can be obtained from improving protein sequence matching itself (Proteins 2021, 89:671), which yields many more of the structure contacts.

The pipeline was tested on 221 non-redundant protein domains taken from scope and FM targets form various CASPs. Results were compared against several other prediction approaches and clear gains were seen.

In the Author Summary should ‘spare’ be ‘sparse’ on next to last line?

Reviewer #2: The authors present DeepFold, a new method for protein structure prediction that uses L-BFGS down-hill minimization of distance and orientation constraints, formulated as an energy function. The constraints are derived from both MSAs and simple physical considerations. A similar approach is implicit in AlphaFold1. The authors carefully analyze the contribution of different constraints, and combinations thereof, to the shape of the energy function and prediction quality, as manifested by the TM-scores of the resulted models. They also provide visual inspection of a few interesting test cases. The new method outperforms several important alternatives in both accuracy and speed. Notably however, in this study DeepFold is not compared with AlphaFold2 and RoseTTAfold, which are freely available for testing.

The major take-home message of this study is that the larger is the number of reliable constraints, and the more diverse they are, that is distance vs. orientation and MSA-based vs. physics-based, the smoother is the energy surface and consequently the more accurate are the predictions.

Overall, the manuscript is clearly written and interesting, with minor issues raised below. However, its major weakness is the lack of comparisons to AlphaFold2 and RoseTTAfold, which are likely to outperform DeepFold. Obviously, this speculation may be wrong, and anyway performance is not the only criterion for the evaluation of a research paper, but the “elephant in the room” here is far too big to ignore.

Minor issues:

1. The authors compare methods and sets of constraints by average values of performance measures (e.g., TM-score), and test the significance of differences of these values by Student’s t-test. Averages and parametric tests maybe misleading when applied to distributions that are not normal like the ones shown in Figure 2. I believe that Instead (or in addition), the authors should use more robust, non-parametric, values and tests.

2. The authors mention “16 targets with Neff values less than 1 “. I guess this is a mistake. Even an orphan has a Neff value of 1.

3. When mentioning the use of Potts models as features, the authors should refer to previous studies that did it. Further, I believe that readers would benefit from some discussion about the meaning of Potts models in this context. If space is limited, we could do without a formal description of a much more known algorithm such as L-BFGS.

Reviewer #3: # General comments

In this work, Pearce and colleagues present DeepFold, a novel method to predict

high-quality protein three dimensional structures using an ab initio approach

which is guided by restraints originating from deep learning.

In my opinion, DeepFold constitutes a significant advance in the field, as the

authors convincingly demonstrate the high quality of results obtained while

maintaining running times reasonable. This performance speed-up is achieved by

exploiting the wealth of structural restraints obtained from deep neural network

processing of sequence-derived information, which enables the application of the

very efficient L-BFGS algorithm in a smoothed energy landscape, leading to fast

convergence.

The manuscript is well written and provides detailed benchmarking to relevant

methods, suggesting that DeepFold can yield high quality results in reasonable

time, thus making DeepFold a new important addition in the toolset of protein

structure prediction.

Below are some specific comments (grouped by manuscript section), which I

hope will be helpful to the authors. In addition, a few typos and potential

passages needing clarifications are mentioned in the end of my review.

# Specific comments

+ Results

- The authors clearly demonstrate (Fig. 4) that DeepFold outperforms competing

methods in the compiled benchmark data. I would find interesting a discussion

on the identities/properties of the few proteins where DeepFold performed worse

than its 'competitors'. Were these inferior predictions due to poor MSAs (e.g.

low Neff) or is there another reason that the authors could identify?

+ Methods and Usage

- I was unable to get results on time for my review using the online server.

However, I trust that the server performs as advertised, as the authors

provide an example input for submission and the corresponding output, which

includes a predicted model for the example query sequence along with the

intermediate results (i.e. predicted secondary structure and spatial restraints).

In addition, the authors have made available a GitHub repository with the DeepFold

source code accompanied with detailed documentation on how to install, setup and

execute the DeepFold suite. Therefore, both regular and more experienced users

will find it easy to get their hands on this new method.

- In the https://zhanggroup.org/DeepFold/README.md file it is mentioned that

"Perl and java interpreters should be installed". It would be helpful if the

minimum required versions for these interpreters are mentioned along with

any non-standard packages/classes on which the provided code is depending.

- In the manuscript the authors mention the use of DeepMSA2 for deriving a

multiple sequence alignment based on the query sequence. However, if I am not

mistaken, the code provided in the repository corresponds to an initial version

(DeepMSA). I would suggest the authors were given the opportunity to choose

which version to install/use.

- In page 13 (last paragraph) qMSA is described and it is mentioned to use

"HHblits2 to search against the Uniclust30 database". However it is unclear

from this passage which is the query used at this point. A reader needs to

consult the respective supplementary figure (Fig S1), where it is shown that

this step is executed using the original query.

In addition, it would be interesting to see the frequency by which each of

the 7 types of MSA are chosen as the final MSA. Such information could provide

additional insights for further speedup of the complete pipeline, since the

MSA construction step requires significant computational resources at all

levels - CPU, main memory, hard disk: potential users interested to install

the code locally (especially in resource limited settings) would be interested

in having information on which parts of this pipeline could be skipped without

affecting quality of the results.

- "DeepPotential models" were trained on a non-redundant set from the PDB.

However, it is unclear by reading the respective section (pg. 14) whether

the protein chains composing this training dataset exhibited any sequence

similarity to the dataset used for benchmarking.

+ Typos/minor clarifications

- Page 9, paragraph 1, line 2: "inter-reside" should read "inter-residue"?

- Page 10, paragraph 2, line 4: I personally disagree with the use of the term

"**highly** statistically significant".

- Page 10, paragraph 3, line 3: Please provide a definition for Neff here, or

give a pointer to the literature for readers not familiar with this term.

- Page 14: "DeepFold Force Field" parameter weights were initialized to zero

values, then optimized. However, it is not mentioned what was the amount of

increase to these weights (I wonder if the weights should be monotonically

increased). The "grid-searching technique" also lacks implementation details.

- Page 19: In Fig 1, at the DeepPotential box, a label "2D input features"

above the 2D residual blocks is probably missing.

**Have the authors made all data and (if applicable) computational code underlying the findings in their manuscript fully available?**

Reviewer #1: **No: **All details of their pipeline have not been included, nor all parameters used - generally these authors have always created publicly available web sites for general open access.

Reviewer #2: Yes

Reviewer #3: Yes

PLOS authors have the option to publish the peer review history of their article (what does this mean?). If published, this will include your full peer review and any attached files.

Reviewer #1: No

Reviewer #2: No

Reviewer #3: **Yes: **Vasilis J Promponas
---

## [Decision Letter · Decision Letter 1]

3 Sep 2022

Dear Dr. Zhang,

We are pleased to inform you that your manuscript 'Fast and Accurate Ab Initio Protein Structure Prediction Using Deep Learning Potentials' has been provisionally accepted for publication in PLOS Computational Biology.

Best regards,

Christos A. Ouzounis

Academic Editor

PLOS Computational Biology

Arne Elofsson

Section Editor

PLOS Computational Biology

Reviewer's Responses to Questions

**Comments to the Authors:**

Reviewer #1: Bravo. I am particularly glad to see your important progress reported here. To me your paper is significantly stronger after responding to all 3 reviews in the way you have!

Reviewer #3: I thank the authors for their efforts to addressed all comments raised in the initial round of review.

I have no further comments.

**Have the authors made all data and (if applicable) computational code underlying the findings in their manuscript fully available?**

Reviewer #1: Yes

Reviewer #3: Yes

PLOS authors have the option to publish the peer review history of their article (what does this mean?). If published, this will include your full peer review and any attached files.

Reviewer #1: No

Reviewer #3: **Yes: **Vasilis J Promponas

---

## [Editor Report · Acceptance letter]

12 Sep 2022

PCOMPBIOL-D-21-02319R1 

Fast and Accurate Ab Initio Protein Structure Prediction Using Deep Learning Potentials

Dear Dr Zhang,

I am pleased to inform you that your manuscript has been formally accepted for publication in PLOS Computational Biology. Your manuscript is now with our production department and you will be notified of the publication date in due course.

With kind regards,

Zsofia Freund
